# Mitochondrial Targeting against Alzheimer’s Disease: Lessons from Hibernation

**DOI:** 10.3390/cells13010012

**Published:** 2023-12-20

**Authors:** Christina F. de Veij Mestdagh, August B. Smit, Robert H. Henning, Ronald E. van Kesteren

**Affiliations:** 1Department of Molecular and Cellular Neurobiology, Center for Neurogenomics and Cognitive Research, Vrije Universiteit Amsterdam, 1081 HV Amsterdam, The Netherlands; guus.smit@vu.nl (A.B.S.); ronald.van.kesteren@vu.nl (R.E.v.K.); 2Department of Clinical Pharmacy and Pharmacology, University Medical Center Groningen, 9713 GZ Groningen, The Netherlands; r.h.henning@umcg.nl; 3Alzheimer Center Amsterdam, Amsterdam UMC Location VUmc, 1081 HV Amsterdam, The Netherlands

**Keywords:** mitochondrial dysfunction, Alzheimer’s disease, daily torpor, hibernation-derived compound, SUL-138

## Abstract

Alzheimer’s disease (AD) is the most common cause of dementia worldwide and yet remains without effective therapy. Amongst the many proposed causes of AD, the mitochondrial cascade hypothesis is gaining attention. Accumulating evidence shows that mitochondrial dysfunction is a driving force behind synaptic dysfunction and cognitive decline in AD patients. However, therapies targeting the mitochondria in AD have proven unsuccessful so far, and out-of-the-box options, such as hibernation-derived mitochondrial mechanisms, may provide valuable new insights. Hibernators uniquely and rapidly alternate between suppression and re-activation of the mitochondria while maintaining a sufficient energy supply and without acquiring ROS damage. Here, we briefly give an overview of mitochondrial dysfunction in AD, how it affects synaptic function, and why mitochondrial targeting in AD has remained unsuccessful so far. We then discuss mitochondria in hibernation and daily torpor in mice, covering current advancements in hibernation-derived mitochondrial targeting strategies. We conclude with new ideas on how hibernation-derived dual mitochondrial targeting of both the ATP and ROS pathways may boost mitochondrial health and induce local synaptic protein translation to increase synaptic function and plasticity. Further exploration of these mechanisms may provide more effective treatment options for AD in the future.

## 1. Introduction

Alzheimer’s disease (AD) is the most prevalent neurodegenerative disease worldwide, and as the population ages, the numbers are only expected to rise [1]. So far, despite decades of research and efforts, no cure has been found for this devastating disease. Treatments targeting either of the two main pathological hallmarks of the AD brain, amyloid-beta (Aβ) and tau aggregation, have been mostly unsuccessful, driving the search for alternative treatable causative agents of AD. A growing body of evidence supports a causal role of mitochondrial dysfunction in AD [2,3,4,5,6]. This has led to the mitochondrial cascade hypothesis of AD, proposing a progressive decrease in mitochondrial fitness as causative factor in AD. Until to now, small molecule compounds with antioxidant (i.e., ROS-scavenging) properties, such as vitamin E and curcumin, have been the most studied as mitochondrial targeted treatments in AD [7,8,9]. While antioxidants show some promising results in in vitro and mouse models of AD, they have failed to restore cognitive function in clinical studies, possibly due to poor blood–brain barrier permeability and the singular effect of ROS scavenging without improving mitochondria function itself [9,10].

Remarkably, hibernating animals show exceptional brain plasticity when alternating between torpor (periods of hypothermia and hypometabolism) and arousal (periods when metabolism and body temperature return to normal) [11,12,13,14,15,16,17,18,19]. Hypometabolism during torpor is caused by the active shutdown of the mitochondria, which is rapidly reversed upon arousal without causing ROS damage. Depending on the species and environmental conditions, seasonal hibernators, such as ground squirrels and dormice, employ deep and long torpor bouts (1–3 weeks typically at ~5 °C), whereas daily hibernators, such as Siberian hamsters and mice, employ short (4–10 h at ~17 °C) torpor bouts [20,21]. The regulation of mitochondrial activity in hibernation coincides with extensive neuronal plasticity in many brain regions, ranging from complete dendritic retraction and restoration in seasonal hibernators to specific synaptic changes during daily torpor in mice [11,22,23]. In addition, daily torpor in mice increases the levels of mitochondrial complex I and IV proteins, proteins involved in synaptic plasticity, long-term potentiation in the hippocampus, and restoring memory function in an AD mouse model [23]. Interestingly, mimicking arousal-induced mitochondrial changes using the 6-chromanol SUL-138, which supports complex I and IV function while suppressing ROS production, was able to mirror these effects [24]. This suggests that hibernation-associated mitochondrial regulation could be beneficial in treating AD-related synaptic and mitochondrial impairments.

In this review, we briefly discuss mitochondrial deficiencies that lead to synaptic impairment in AD and mitochondrial targeted intervention strategies that have so far proven insufficient in improving AD clinical outcomes. Next, we discuss mitochondria in hibernation and how hibernation-derived mitochondrial targeting offers new and unique opportunities in the search for novel treatment strategies against AD, including current advances. Finally, we propose a model through which hibernation-associated mitochondrial activation could lead to increased synaptic plasticity relevant for the treatment of cognitive decline in AD.

## 2. Mitochondria in AD

### 2.1. Mitochondrial Dysfunction in AD: The Mitochondrial Cascade Hypothesis

The mitochondrial cascade hypothesis, which originated in 2004 [25], is gaining attention, and by now, a large body of evidence implicates mitochondrial dysfunction as a fundamental causative mechanism in AD [6,26]. The hypothesis states that a progressive decrease in mitochondrial fitness, excess release of reactive oxygen species (ROS), and decreased ATP production are important contributors to pathological protein aggregation, decreased synaptic plasticity, and cognitive decline in AD.

Already over 40 years ago, electron microscopy and PET imaging revealed altered mitochondrial morphology and decreased glucose metabolism in AD brains [27], a parameter that appears superior to tau or Aβ pathology in predicting AD progression [28]. Mitochondrial deficits, as reflected in increased oxidative stress (e.g., free radical production, lipid peroxidation, oxidative protein damage) [29] and altered calcium homeostasis [30], are observed in AD patients prior to histological and clinical abnormalities [31,32,33]. By now, it is clear that the mitochondria are differentially affected in early- vs. late-stage AD and that neurons and glia may be differentially affected during disease progression, providing new opportunities for early diagnosis and intervention [34].

Interestingly, caloric restriction, intermittent fasting, and ketogenic diets that alter metabolic input to mitochondria have shown mild beneficial effects in animal models of neurodegenerative disorders and human clinical trials [35,36]. Consequently, mitochondria should be seriously considered as potential pharmacotherapeutic targets in the treatment of AD.

### 2.2. Glucose Dysmetabolism and Oxidative Stress: An ATP/ROS Disbalance in AD

The brain is the organ with the highest energy demand, and therefore, it relies heavily on glucose (the predominant substrate for the human adult brain under physiological conditions) to efficiently produce ATP. The production of ATP occurs through glycolysis, the tricarboxylic acid (TCA) cycle, and the oxidative phosphorylation system (OXPHOS) via the electron transport chain (ETC) [6]. Neurons are mostly oxidative, and glial cells process glucose glycolytically, with the astrocyte–neuron lactate shuttle metabolically coupling the different cell types [37]. The impairment of glucose metabolism in AD brains is detrimental for the ATP levels necessary for normal functioning of neurons [29,38,39,40,41]. A main component in the dysregulation of glucose metabolism is oxidative stress caused by ROS, an inevitable, albeit normally well-handled, by-product of oxidative phosphorylation, produced mostly by complexes I and III. Mitochondrial dysfunction caused, e.g., by aging accumulated damage or genetic defects, leads to increased ROS production, causing oxidative modification of the enzymes involved in glycolysis and the TCA cycle, reducing their efficiency and creating a negative feedback loop that further impairs mitochondrial function [6,29,42] (Figure 1). In addition, many studies have shown reduced levels of all five complexes of the ETC in AD brains, leading to further impairment of ATP production [3,43,44,45,46,47]. Moreover, the efficiency of all five complexes is lowered in AD, with a particularly clear defect in cytochrome c oxidase (complex IV), leading to increased ROS levels and further ATP deficiency [48].

### 2.3. Mitochondrial Impairment Leading to Synaptic Dysfunction in AD

Whether or not the mitochondrion represents a true causative agent from which AD originates, mitochondrial dysfunction plays an eminent role in synaptic impairment in AD [49]. Synaptic dysfunction is an early pathological feature and correlates highly with memory loss in AD [50,51,52]. Neurons heavily rely on healthy functioning mitochondria to meet the high energy demands, e.g., for mobilizing synaptic vesicles, synaptic transmission, maintaining ion gradients for action potential generation, and propagation and local mRNA translation for synapse function and synaptic plasticity [6,49]. Decreased *ATP* levels in AD brains can interfere with each of these processes [53,54,55,56]. In addition, increased *ROS* levels in AD brains can lead to impaired mitochondrial Ca^2+^ buffering, which in turn, leads to the loss of the microtubule assembly-dependent transport of synaptic vesicles and mitochondria, further lowering the energy supply to synapses and additionally impairing synaptic transmission [57,58].

### 2.4. Mitochondrial Targeted Strategies in AD

Drugs that effectively target Aβ or Tau pathological protein aggregation have not lived up to their therapeutic promise in AD yet. In the context of the mitochondrial cascade hypothesis, therapeutics targeting mitochondria have been tested both in preclinical and clinical settings. As such, antioxidants are obvious candidates to mitigate ROS damage [29]. Treatment with antioxidants, such as Vitamin E and C [8,59,60] and Ginkgo biloba extract [61,62], a natural antioxidant extensively used in Chinese traditional medicine, have not entered the AD drug market yet due to limited effects in patients. However, recent studies do show mildly positive effects for Ginkgo biloba extract [63,64], but the final call on these treatments has not been made yet. Another group of mitochondrial compounds that have been shown to counteract mitochondrial dysfunction and oxidative stress are phenylpropanoids [65,66,67]. For example, curcumin, a commonly known spice, showed promising results on oxidative damage and Aβ plaque burden in an APP/PS1 AD mouse model [68]. However, positive clinical outcomes in humans remain absent, presumably due to low bioavailability [69,70]. Apart from relieving ROS levels, stimulating or protecting mitochondrial ATP production has been attempted using, for example, oxaloacetate (OOA) [71], an intermediate of the TCA cycle, or precursors of nicotinamide adenine dinucleotide (NAD) [72], an intermediate common to several mitochondrial metabolic pathways, including glycolysis, TCA cycle, and oxidative phosphorylation, but clinical efficacies still remain to be studied [65]. Interestingly, mild inhibition of complex I of the OXPHOS system, e.g., by the small molecule tricyclic pyrone compound CP2, has been suggested as a potential treatment strategy against AD, with positive effects in the APP/PS1 mouse model of AD [73,74]. Likewise, the inhibition of complex V (ATP synthase) using the curcumin derivative J147 preserves mitochondrial homeostasis and prevents cognitive decline in SAMP8 mice [75,76]. It is thought that the inhibition of complex I, while lowering ATP production, exerts its beneficial effects by triggering a mitochondria-mediated integrated stress response and due to lower ROS production via complex I [72] However, clinical studies with non-specific complex I inhibitors, such as metformin and resveratrol, have been unsuccessful, possibly due to limited bioavailability and non-specificity [77,78]. Although the general consensus is that mitochondria should be seriously considered as targets against AD, issues with bioavailability, specificity, efficacy, and side effects of mitochondrial targeting have hindered progression in this field beyond preclinical findings. Thus, novel strategies and out-of-the-box approaches are needed to overcome this hurdle.

## 3. Mitochondria in Hibernation

### 3.1. Metabolic Adaptation during Hibernation

Hibernation is a natural phenomenon during which animals escape energetically challenging environmental conditions by entering torpor, a state of extreme hypometabolism and hypothermia. During torpor, hibernators show a drastic, up to 98%, reduction in metabolic rate [79]. Though its exact molecular underpinnings are still poorly understood, torpor features the halt of glycolysis, which is established by changing the levels and activities of several key enzymes involved in metabolic conversion during glycolysis, pyruvate metabolism, and the TCA cycle, including glycogen phosphorylase, pyruvate dehydrogenase, phosphofructokinase, pyruvate kinase, and citrate synthase [79,80,81,82,83,84] (Figure 2). ATP production is further halted due to a reduction in the efficiency (but not levels) of complexes I and II [85] of the OXPHOS and of the mitochondrial H_2_S oxidizing enzyme sulfide: quinone oxidoreductase (SQR), which provides electrons to the ETC via coenzyme Q [86,87]. The activities of these OXPHOS proteins are thought to be regulated though allosteric inhibition by oxaloacetate, a TCA cycle intermediate that accumulates due to decreased activity of TCA cycle enzymes, and through posttranslational modifications by intramitochondrial kinases and deacetylases [88]. In addition, reduced efficiency of H_2_S-oxidizing SQR and increased levels of H_2_S-synthesizing enzymes in the brain, mainly cystathionine beta-synthase (CBS), leads to an increase in H_2_S levels during torpor, which inhibit complex IV (cytochrome c oxidase) function of the OXPHOS system [89]. The limited remaining metabolism during torpor shifts to fatty acid metabolism [81,83,90]. During arousal, reactivation of the mitochondria occurs in a matter of hours. This reactivation initially leads to ‘non-shivering thermogenesis’ via ATPase uncoupling in brown adipose tissue (BAT). Further rewarming is caused as a by-product of reinstated ATP production in other tissues, such as skeletal muscle (‘shivering thermogenesis’). Lipids are the predominant fuel during the high-energy demanding arousal phase [79,83,84,91]; however, other non-lipid fuels, such as carbohydrates and glycogen, have also been shown to be recruited upon arousal in 13-lined ground squirrels [92]. In addition, changing mitochondrial architecture during hibernation through fission and fusion has been suggested as a potential regulator of metabolism during hibernation [93]. Even though gaps in our knowledge on mitochondrial adjustments during hibernation remain, the abovementioned mechanisms involved in healthy mitochondrial regulation might harbor new AD treatment options.

### 3.2. Defense Mechanisms against ROS Damage during Hibernation

Interestingly, the suppression and re-activation of mitochondrial function during torpor and arousal occurs while maintaining sufficient energy supply and without the occurrence of ROS damage normally seen during rewarming and reperfusion [84,94]. Excess substrate at—and reverse electron flux through—complex II during ischemia–reperfusion is usually a significant source of electrons for ROS production at complex I [84,85]. Therefore, protection against ROS production during arousal has been linked to the significant reduction in complex I and II function during torpor [95]. In addition, brown adipose mitochondria, which uncouple respiration from ATP for non-shivering thermogenesis and release the energy of the proton gradient directly as heat, are also protected against ROS damage. This respiratory uncoupling of BAT mitochondria is due to relatively high levels of uncoupling protein 1 (UCP1), which reduces proton motive force and is thought to alter the redox state of the respiratory chain, effectively reducing the production of ROS [96]. Other putative oxidative stress protective strategies associated with hibernation are the increased levels of antioxidants, such as ascorbate and glutathione during arousal [90,97].

ROS damage preventing strategies observed during hibernation have been of particular interest for the ischemia–reperfusion field, as, in humans, the overproduction of reactive oxygen species (ROS) upon reperfusion jeopardizes cellular integrity [98,99]. Further exploiting the mechanisms that hibernators use to maintain sufficient mitochondrial ATP production without ROS damage may thus also represent a promising therapeutic strategy in AD, given that mitochondrial dysfunction and subsequent ATP/ROS imbalance are prominent in the AD brain.

### 3.3. Mitochondria in Daily Torpor Mice

Though long overlooked, laboratory mice are also capable of entering a hibernation state, particularly daily torpor [23,100,101,102,103,104]. They exert facultative daily torpor as a response to energetic challenges, such as low food availability or increased energy expenditure. Torpor in mice is generally induced using a fasting protocol, completely omitting food availability, or a work-for-food protocol, in which mice have so-called high foraging-costs, typically due to wheel running as a requirement to access food [105,106]. Both ultimately lead to a negative energy balance that is compensated for by daily torpor cycles with body temperatures as low as 21 °C and a metabolic rate reduction up to 70%. It has been found that, like seasonal hibernators, the efficiency of complexes I and II of the OXPHOS system and of ADP phosphorylation is reduced during daily torpor in Balb/c, CD1, and C57BL/6N mice [95]. In addition, we observed robust hippocampal mitochondrial protein regulation during daily torpor in C57BL/6 mice [23]. Notably, many of these mitochondrial proteins were part of electron transport chain complexes I and IV, previously shown to be involved in mitochondrial activity regulation during hibernation [79,84,85,88]. The effects of arousal on mitochondrial protein levels are in line with a study showing that leptin, which is reduced during torpor and reinstated during arousal, normally regulates several OXPHOS proteins [102,107]. It would be interesting to additionally assess complex activity in daily torpor mice in comparison to parallel data from previous studies on mitochondrial regulation during hibernation.

## 4. Hibernation-Derived Mechanisms of Mitochondrial Targeting: Is It Relevant for the Treatment of AD?

### 4.1. Targeting UCP1

The significance of hibernation-derived mitochondrial activation in neuronal protection has been demonstrated in a study that compared iPSC-derived neurons from the hibernating arctic squirrel to iPSC-derived neurons from humans during cold exposure. They found that neurons derived from the arctic squirrel exerted cell autonomous protection against cold-induced stress, which normally results in ROS overproduction, lysosomal membrane permeabilization, and consequential microtubule destruction. This neuronal protection seems to be conferred via torpor-associated mitochondrial uncoupling [108], as mimicking this effect using the mitochondrial uncoupler BAM15 was able to protect against cold-induced stress in human iPSC-derived neurons. The transient overexpression of mitochondrial uncoupling proteins UCP1 or UCP2 likewise produced cold-stable neurites supporting the theory that the protective effects of BAM15 indeed rely on mitochondrial uncoupling. BAM15 has also been tested in a *C. elegans* model of Alzheimer’s disease as Alzheimer’s disease could also benefit from this neuronal protection from ROS damage. Indeed, BAM15 was able to relieve neurodegeneration and aging in this model, as it reduced abnormal shapes of mechanosensory neuronal cells and maintained touch-response and short-term memory [109]. However, further studies are needed to substantiate causal effects of mitochondrial uncoupling on neuronal protection, and relevance for AD in other more complex AD models. The notion that mitochondrial uncoupling can protect neurons from ROS damage and devastating microtubule destabilization that normally occurs during cold-induced stress, is in line with the limited damage that occurs during extreme hypothermia in torpor, which is paralleled by a strong upregulation of UCP1 in brown adipose tissue mitochondria and neuronal mitochondria during this phase [110].

### 4.2. Targeting Mitochondrial Ca^2+^ Handling

Alterations in mitochondrial calcium homeostasis during hibernation confers another benefit of hibernation that could be interesting for AD treatment. Intracellular calcium handling is important for regulating membrane excitability, signal transduction, neurotransmitter release, synaptic plasticity, cell cycle, cell migration, and axon growth. It is regulated through plasma membrane channels and intracellular storage in the endoplasmic reticulum, Golgi apparatus, and mitochondria. Disrupted Ca^2+^ homeostasis, e.g., following NMDA receptor activation, leads to mitochondrial dysfunction and to rapidly increasing mitochondrial Ca^2+^ levels, resulting in increased mitochondrial membrane depolarization and excitotoxic cell death [30,111,112,113]. During hibernation, the mitochondrial calcium uniporter (MCU) complex, important for transporting calcium into the mitochondrion, and leucine zipper and EF-hand containing transmembrane protein 1 (LETM1), which mediates calcium release from the mitochondrion, are upregulated in skeletal muscle in Daurian ground squirrels [114,115]. It is thought that the regulation of mitochondrial calcium handling via these complexes helps hibernators to maintain intracellular Ca^2+^ homeostasis during the extremes of hibernation, for example, in the heart and skeletal muscle, which, otherwise, would suffer from cytoplasmic Ca^2+^ overload during inactivity [114,116]. In addition, it has been shown that a high rate of mitochondrial Ca^2+^ uptake lowers the OXPHOS proton-motive force sufficiently to cause a transient reversal of the ATP synthase, which is necessary for ATP synthesis shutdown during torpor. Studies on Ca^2+^ homeostasis in the brain tissue of hibernators are scarce; however, one study reported reduced intracellular Ca^2+^ levels in ground squirrel cerebral synaptosomes during torpor, which is thought to increase tolerance against prolonged ischemia during hibernation [117]. Calcium dyshomeostasis in AD leads to intracellular calcium overload, resulting in neurodegeneration via its effects on mitochondrial- and synaptic dysfunction and affecting the production and aggregation of Aβ peptides and tau hyperphosphorylation [118]. Therefore, several studies have suggested the potential of reducing mitochondrial Ca^2+^ uptake as a potential treatment strategy against AD. MCU deletion improved mitochondrial function and prevented neurodegeneration in *C. elegans* expressing AD relevant mutations in their PSEN homolog, sel-12 [119], and in a 3xTG mouse model of AD [120]. In addition, a knockdown of endogenous MCU rendered primary cultured neurons resistant to oxidative stress [121], and the MCU inhibitor Ru360 prevented mitochondrial Ca^2+^ uptake in mice in vivo after exposure to soluble Aβ [122]. However, all these studies focused on inhibiting MCU activity, thereby solely preventing Ca^2+^ influx into the mitochondrion. Yet, hibernators upregulate both MCU and LETM1, stimulating both Ca^2+^ import and export, possibly to enhance the overall calcium buffering capacity of the mitochondrion. Further studies should explore this dual targeting of mitochondrial calcium handling as a neuroprotective strategy in AD.

### 4.3. Targeting H_2_S

H_2_S is a gasotransmitter, similar to NO and CO, and regulates numerous cytoprotective and physiological functions through its anti-oxidative and anti-inflammatory actions [123]. Interestingly, in 2005, researchers found that low doses of H_2_S inhalation led to suspended animation (a torpor-like state) in mice [124]. Though questions remain regarding the low O_2_ levels used in this study (17.5%, which is hypoxic) [94], it sparked interest in the role of H_2_S in hibernation, and it is now thought that H_2_S plays a key role in suppressing mitochondrial activity during torpor [87]. As previously mentioned, the level of H_2_S is regulated through SQR and CBS, and H_2_S inhibits complex IV function by binding Fe^2+^ in heme. The role of H_2_S in AD has also been gaining attention over the years. AD patients show lower levels and reduced activity of the H_2_S-producing enzyme CBS and lower levels of H_2_S in the brain, which correlate to AD severity [125,126,127]. Increasing H_2_S levels as a therapeutic strategy against AD has been extensively reviewed [128,129]. Indeed, increasing H_2_S levels similar to what is seen during hibernation leads to the relief of AD-related pathophysiologic outcomes in various animal and cell models of AD. For example, NaHS, a H_2_S donor, ameliorated Aβ-induced damage in PC12 cells by reducing the loss of mitochondrial membrane potential and attenuating the increase in intracellular ROS [128]. In addition, increasing H_2_S levels in the brain of AD animal models via a donor or through a diet rich in taurine, cysteine, folate, B12, and betaine, has been shown to be beneficial against oxidative stress, neurodegeneration and cognitive impairment [129,130]. However, the clinical translation of these results is hindered by the complex dosage regimen of a H_2_S donor, requiring multiple high doses, and H_2_S’s potential toxicity at such high doses [129]. Currently, an H_2_S-targeting compound, the CBS activator s-adenosyl methionine (SAM), has entered a phase II clinical trial, studying its effects in patients with mild cognitive impairment. The results will hopefully shed more light on the potential of H_2_S-based treatments in AD patients [131].

### 4.4. Targeting Complex I or II

Recently, the mRNA expression levels of OXPHOS subunits, primarily complex II, were shown to be decreased during torpor in 13-lined ground squirrels, while the levels of TCA cycle intermediates prior to complex II were simultaneously increased, suggesting that hibernation halts the TCA cycle input towards OXPHOS through complex II inhibition [132]. This is thought to prevent ROS formation during hibernation via blocking reverse electron transport from complex II to complex I. Mimicking this effect via blocking complex II with dimethyl malonate (DMM) was able to rescue hypoxia damage in hypoxic human SH-SY5Y differentiated neurons and in vivo in ischemic stroke in mice [132]. In addition to complex II inhibition, the inhibition of complex I, which is also downregulated during hibernation, has been suggested as a potential strategy against AD as well, with positive effects in an APP/PS1 mouse model of AD [73,75]. However, as mentioned previously, clinical studies with non-specific complex I inhibitors, such as metformin and resveratrol, were not successful [77,78]. Solely targeting ROS damage by mimicking torpid mitochondria without stimulating ATP production might, therefore, be too unidirectional in the treatment of AD.

### 4.5. Dual Targeting: Improving ATP/ROS Balance

The notion that laboratory mice are also capable of daily torpor [23,100,102,103,104,133,134,135], showing bouts of torpor and arousal of several hours instead of days to weeks, offers the opportunity to explore hibernation in a standard animal model and in a relatively short timeframe, with AD mouse models readily available to study its relevance for AD. We found an upregulation of mitochondrial complex I and IV proteins during arousal after torpor in mice, which was paralleled by a significant increase in the levels of proteins involved in synaptic plasticity, such as AMPAR/NMDA receptor subunits, CAMK2A, and SHISA6/7, and in long-term potentiation and learning and memory [23]. Interestingly, a single torpor bout was sufficient to rescue memory impairment in an APP/PS1 AD mouse model. These observations suggest that arousal-induced mitochondrial activation has the potential to enhance synaptic transmission and plasticity. Therefore, hibernation-derived molecules, which are based on the protective mechanisms that hibernators utilize to withstand damage occurring during the repeated cycles of torpor and arousal, might be advantageous in treating AD. The 6-chromanol derivative SUL-138 was selected as a lead compound through the screening of a small molecule library in a hypothermia-rewarming (H/R) cell model. The mode of action of SUL-138 constitutes mitochondrial complex I and IV activity preservation under pathological conditions [24], resulting in the maintenance of tissue energy levels (i.e., ATP) and a reduction in oxidative stress (i.e., ROS production and lipid peroxidation), thereby mimicking the arousal-associated mitochondrial state (Figure 3). Three months of the oral administration of SUL-138 increased synaptic transmission and memory performance in both APP/PS1 and wildtype mice, similar to a single torpor/arousal cycle [24]. Notably, improved memory in SUL-138-treated APP/PS1 mice was accompanied by a partial rescue of synaptic protein expression, and a significant upregulation of mitochondrial proteins involved in fatty acid degradation and oxidation and a substantial decrease in brain amyloid plaque load and size. Collectively, these data suggest that SUL-138 not only activates complexes I and IV but also alters metabolic input in the mitochondrion, possibly further enhancing ATP production while limiting ROS. However, direct measurements of mitochondrial respiration are needed to confirm this. The SUL-138-evoked long-term metabolic adaptations in mitochondria, and its effects on synaptic transmission and memory performance, illustrate that targeting mitochondrial bioenergetics might be a promising strategy to prevent cognitive impairment in AD. Interestingly, SUL-138 supports complex I and IV activity without affecting basal mitochondrial membrane potential or causing apparent mitochondrial toxicity [136]. Moreover, hibernation-inspired 6-chromanols prevent organ damage in various preclinical models of conditions or diseases with impaired mitochondrial function, including whole body cooling [137], renal ischemia/reperfusion [136], COPD [138,139], and diabetes [140]. As SUL-138 is permeable through the blood–brain barrier, it would be a suitable candidate to also treat AD. In this regard, it is important to further explore how dual hibernation-derived mitochondrial targeting would be able to increase synaptic plasticity in AD.

## 5. Hibernation-Derived Mitochondrial Reactivation and Synaptic Plasticity: A Possible Link with Relevance for AD Treatment

As discussed, hibernation affects both mitochondrial function and synaptic plasticity, and it is interesting to speculate how the two may be linked. LTP and other fast-acting postsynaptic plasticity processes are often facilitated by increased synaptic protein translation, a process that is impaired in neurodegenerative diseases, such as AD [54,141,142,143]. We postulate that mitochondrial priming increases synaptic de novo protein translation of postsynaptic proteins, which in turn, increases LTP and memory formation (Figure 4), and that mitochondrial dysfunction in AD directly impacts synaptic function and cognition. Future studies should aim to substantiate this, e.g., by measuring translation in isolated torpor/arousal and SUL-138-treated synaptoneurosomes [144,145,146] as a measure of synaptic local de novo protein translation, and by the short-term selective inhibition of complex I and IV activation during arousal or SUL-138 treatment, while assessing local translation, synaptic plasticity, and memory.

## 6. Limitations of Translating Mitochondrial Adaptations in Hibernation to AD

Although increasing evidence points to a significant role of mitochondrial dysfunction in AD [26,147,148], the role of amyloid plaques and hyperphosphorylated tau tangles as pathological actors in the disease should not be ignored. We found that a 3-month treatment with a hibernation-derived mitochondrial targeting SUL compound led to a significant reduction of amyloid plaque size and numbers in APP/PS1 mice [24], indicating that targeting mitochondria can reduce amyloid burden. However, the exact mechanism of action remains unclear. The effects could be explained by either a direct effect on increased mitochondrial health of amyloid deposition or clearance, or by currently unknown indirect effects of the SUL compound on, e.g., enzymes acting on amyloid production or clearance. Future studies should focus on demonstrating a causal role of mitochondrial targeting in decreasing plaque load, for instance, by applying complex I and IV inhibitors during SUL treatment. In addition, the effects of the SUL compound on tau pathology should be investigated in appropriate cell or animal models.

Previous experience has taught that translating therapeutic strategies that show promising effects in AD mouse models to human treatment is often unsuccessful [149,150]. This might be due to AD mouse models mirroring only limited aspects of human AD pathology. Primate models or human iPSC-derived cell cultures may be more suitable and could be used to further substantiate the therapeutic value of hibernation-derived mitochondrial targeting against AD.

Finally, the complex etiology of AD prompts multitargeted therapeutic approaches [31], especially as we currently only diagnose AD in later stages of disease progression, where the disease already affects many different systems. The combined targeting of, for example, amyloid-beta, tau, and mitochondria, might improve treatment efficacy [151]. This would ultimately also lead to personalized treatment for a disease that is increasingly recognized as being heterogenic in its pathology and clinical outcome [152,153].

## 7. Conclusions

The beneficial effects of dual targeting interventions, improving the fitness of mitochondria not only by reducing ROS but also via stimulating ATP production, offers new opportunities in the search for novel AD treatment strategies. This adds to various studies showing advantageous mitochondrial mechanisms employed by hibernation, with relevance for AD treatments but with limited clinical efficacy so far. How mitochondrial (re)activation enhances synaptic plasticity and reduces AD pathology in AD mice remains to be explored. We suggest enhanced synaptic plasticity, at least in part, relies on the stimulation of the local translation of synaptic plasticity proteins. Altogether, nature’s solution to maintain mitochondrial health in energetically unfavorable conditions may hold substantial promise for the future of AD treatment. 

## Figures and Tables

**Figure 1 cells-13-00012-f001:**
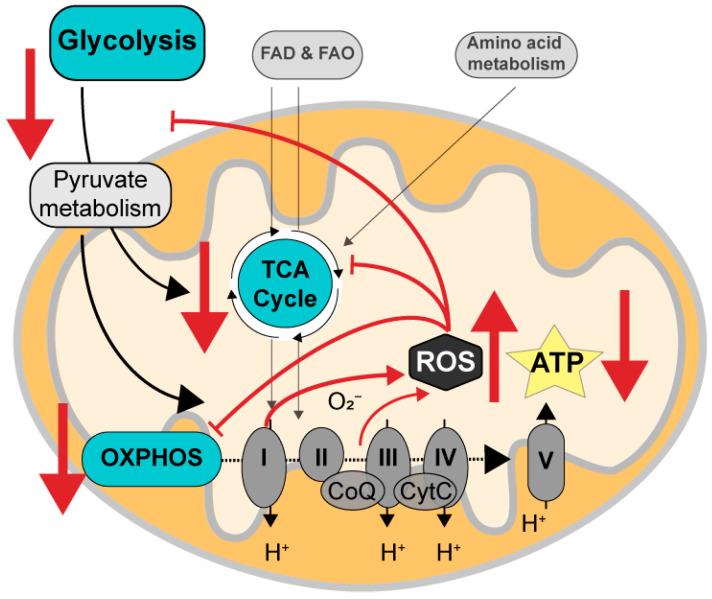
Mitochondrial impairments in AD. The major energy source for the brain is glucose, which is metabolized to ATP via glycolysis, pyruvate metabolism, the tricarboxylic acid (TCA) cycle, and the oxidative phosphorylation system (OXPHOS) via complexes I–V in the electron transport chain. Mitochondria also produce ROS as by-products of activity in the OXPHOS system. Other, less prominent inputs into the OXPHOS system are via fatty acid degradation and oxidation (FAD and FAO) and amino acid metabolism. In AD, excess levels of ROS, probably due to aging-related generation from complexes I and III, leads to oxidative damage to enzymes involved in glycolysis/pyruvate metabolism, the TCA cycle, and the OXPHOS system, thus augmenting ATP deficits, and ultimately leading to high ROS and low ATP levels found in AD (red arrows).

**Figure 2 cells-13-00012-f002:**
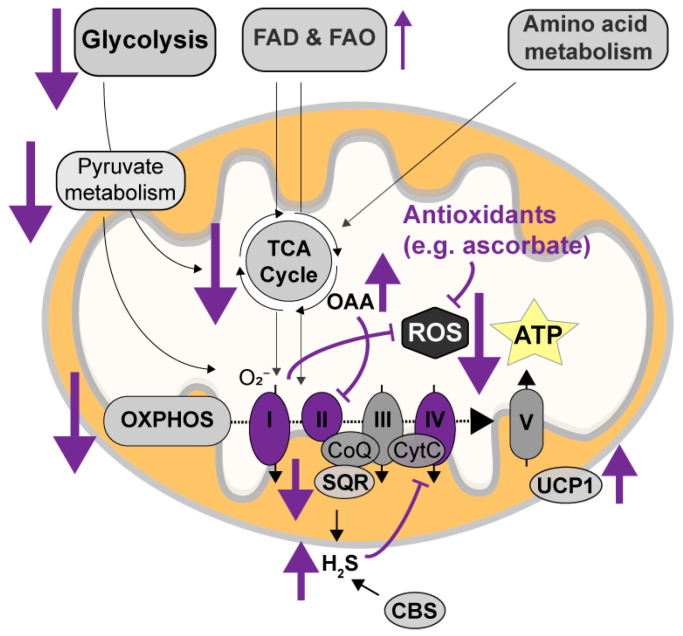
Mitochondrial regulation during hibernation. During the torpor phase of hibernation, glycolysis is halted and remaining energy metabolism shifts to fatty acid metabolism (FAD and FAO) (purple arrows). Enzymes involved in glycolysis, pyruvate metabolism, and the TCA cycle are reduced and/or less efficient. Efficiencies of complexes I and II of the OXPHOS system are drastically reduced through reduced input from the TCA cycle, and posttranslational modifications and allosteric hindrance by oxaloacetate (OAA), an accumulated TCA cycle intermediate. Increased levels of H_2_S due to decreased H_2_S oxidation efficiency of SQR and increased production of H_2_S by CBS, in turn inhibit complex IV function. ROS damage is prevented due to the downregulation of complex I, which normally produces a large fraction of ROS in the OXPHOS system. In addition, UCP1 upregulation leads to uncoupling of the OXPHOS system, thereby altering its redox state and inhibiting ROS production. Finally, antioxidant levels, e.g., ascorbate, are higher during hibernation, directly neutralizing ROS.

**Figure 3 cells-13-00012-f003:**
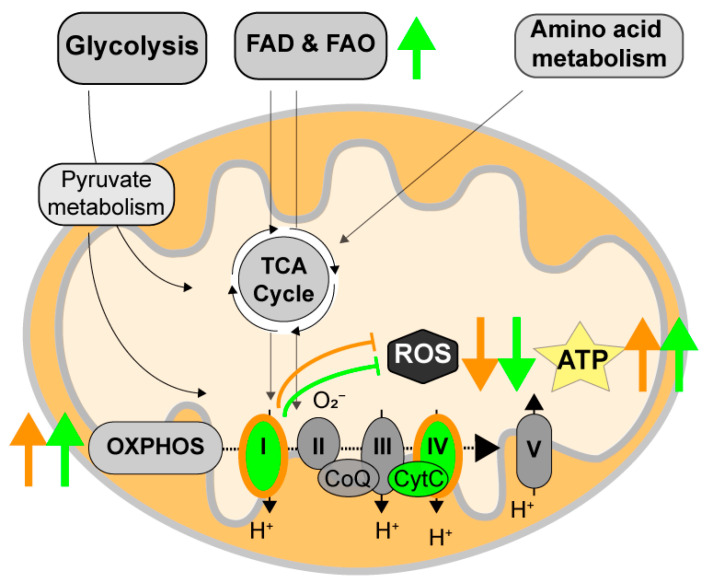
Targeting ATP/ROS balance via mitochondrial stimulation with SUL-138. During arousal from torpor, complexes I and IV are upregulated, and while ROS remains low, ATP levels are reinstated (orange arrows). The 6-chromanol SUL-138 mimics arousal mitochondria after daily torpor in mice by stimulating complex I and IV function, thereby reducing ROS levels and increasing ATP levels. SUL-138 affects complex IV function via the reduction of cytochrome c. In addition, SUL-138 changes metabolic input towards the oxidative phosphorylation system (OXPHOS) to fatty acid degradation and oxidation (FAD and FAO) (green arrows).

**Figure 4 cells-13-00012-f004:**
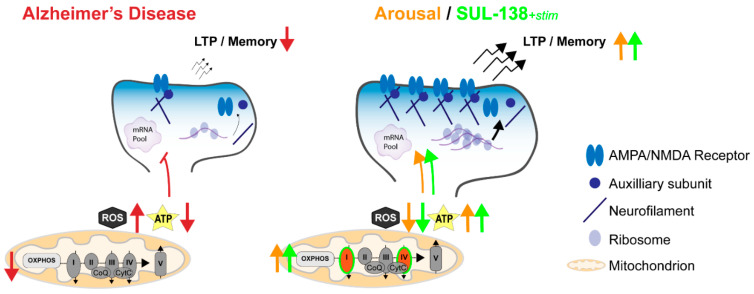
Proposed model for beneficial effects of hibernation-derived mitochondrial activation on synaptic plasticity. Alzheimer’s disease mitochondria have an imbalance in ROS/ATP, with excess ROS production and reduced ATP levels. This leads to pathological secondary outcomes, including inhibited local translation of synaptic plasticity proteins, which depends heavily on sufficient ATP supply. This local translation is important for synaptic maturation, which is essential for LTP and memory, both impaired in AD (red arrows). Mitochondrial reactivation during arousal after torpor offers a unique state in which ROS formation is inhibited and ATP production is reactivated. In daily torpor mice, this arousal phase is characterized by an increase in complex I and IV levels (orange arrows). SUL-138 mimics this reactivation of mitochondria by stimulating complex I and IV function, while preventing ROS formation (green arrows). Therefore, both torpor and torpor-derived mitochondrial activation by SUL-138 can lead to the enhanced local translation of synaptic plasticity proteins (e.g., AMPA/NMDA receptor subunits, auxiliary subunits, and neurofilaments), and hence, may lead to enhanced LTP and memory formation capacity.

## Data Availability

Not applicable.

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
