# Peer review of "Mitochondrial Targeting against Alzheimer’s Disease: Lessons from Hibernation"

_cells, 2023, doi:10.3390/cells13010012_

Round 1

Reviewer 1 Report

Comments and Suggestions for Authors

This review hypothesized that mechanisms acting on animals undergoing hibernation or torpor bouts and that appear to be associated with both decreased ROS production, decreased activi9ty of the respiratory chain followed by its increase during the resolution of torpor bouts, could be exploited fo mitigating AD. AD is a condition invariably associated with amyloidotic plaques due to condensation of Amyloid beta fibrils and hyper-phosphorylated tau tangles. The main pathological consequence of these protein alterations is neuronal loss. A role of mitochondrial decline, possibly linked to ageing, has been hypothesized but the evidence for a prominent dysfunction of mitochondrial bioenergetics in the pathogenesis of AD is controversial. The discussion about adaptive functions of mitochondrial metabolism during hibernation and arousal are interesting but the connection with AD onset and progression is scarce and highly speculative in the absence of robust experimental demonstrations. 

Author Response

The connection of hibernation-derived mitochondrial targeting to AD onset and progression comes from our own work where we were able for the first time to study the consequences of hibernation and hibernation-derived intervention in a mouse model of AD (De Veij Mestdagh et al, Scientific Reports, 2021 and De Veij Mestdagh et al, Alzheimer’s Research and Therapy, 2022). We agree that evidence linking mitochondrial bioenergetics during hibernation to AD onset and progression is scarce and currently only based on mouse studies by a single lab. But we also think that these findings are interesting enough to share in the context of this review, even if the field still is limited (as reviewer 2 acknowledges). We agree with the reviewer that the review should also discuss the limitations of hibernation-derived mitochondrial targeting as therapeutic strategy in AD, and that amyloid beta and tau pathology should not be disregarded. In addition, causal experiments showing the effects of hibernation-derived mitochondrial targeting on AD are currently still missing. We have therefore added a paragraph discussing these limitations of translating mitochondrial adaptations in hibernation to AD (lines 425-450).

Reviewer 2 Report

Comments and Suggestions for Authors

The provided manuscript by de Veij Mestdagh and colleagues provides a very interesting perspective to the somehow exhausted field of Alzehimer's disease therapy. The perspective of learning from hibernation and torpor about re-assessing mitochondrial function is fascinating and might really lead to novel therapeutic approaches. The manuscript is well-written, easy to follow and relies on appropriate literature - even if the field still is limited. I only have minor points to adress:

1) Spelling should be consistent throughout, e.g. complex I or complex-I.

2) There are some minor language issues that would need to be fixed. For example, line 25 increase instead of increases. Or figure titles of Fig. 22 and 4 should start with upper case.

3) Reference 1 might not be the best option to report on actual AD case numbers. The paper by Mangialasche is an excellent contribution to evaluation of therapies, however it is from 2010. Other sources are better to describe patient number development.

4) Just a question: how can torpor be distinguished from caloric restriction as it is induced in lab mice via fasting? I mean - sirtuin expression is also regulated by caloric restriction and thought to be positively acting on AD...

5) Maybe the authors can add a bit more details in regard to mitochondrial dysfunction and AD: is the dysfunction also occuring in prodromal AD/ MCI? Is it similar in early or late onset AD? Is mitochondrial failure restricted to neurons or also occurs in glial cells?

6) The authors state that glucose is the main source of energy for the brain. Potentially, they could at least shortly also mention the lactate-shuttle between astrocytes and neurons (e.g., Magistretti and allaman, Nat Rev Neurosci 2018)?

7) I would ask the authors to be a bit more careful about stating that e.g. Ginkgo biloba extract has proven ineffectiv. There are some recent metaanlayses that hint at a positive effect (Xie eta l., Cells 2022, Fan et al., JAD 2022). The same might be true for the other mentioned compounds. I would ask to maybe give a more balanced view on this.

8) The authors describe very well what happens on the functional level with mitochondria in hibernation. Is there anything known about number of mitochondria or fusion/ fission behavior?

Comments on the Quality of English Language

As stated above, I identified only minor language issues, which should be corrected.

Author Response

1) Spelling should be consistent throughout, e.g. complex I or complex-I.

Spelling consistency was checked throughout and corrected.

2) There are some minor language issues that would need to be fixed. For example, line 25 increase instead of increases. Or figure titles of Fig. 22 and 4 should start with upper case.

All language issues are addresses.

3) Reference 1 might not be the best option to report on actual AD case numbers. The paper by Mangialasche is an excellent contribution to evaluation of therapies, however it is from 2010. Other sources are better to describe patient number development.

Reference 1 is replaced with the 2023 Alzheimer’s disease facts and figures.

4) Just a question: how can torpor be distinguished from caloric restriction as it is induced in lab mice via fasting? I mean - sirtuin expression is also regulated by caloric restriction and thought to be positively acting on AD...

Our experiments included a metabolic control group. 10% of our mice did not enter torpor after the same period of fasting, and these mice did not show any beneficial effects on synaptic plasticity due to caloric restriction only. In addition, during fear conditioning, we controlled for caloric restriction by fasting all mice, and refeeding the control mice shortly before torpor entry. These mice also did not show similar beneficial effects on memory as the torpor mice. Details can be found in the original paper (De Veij Mestdagh et al, Scientific Reports, 2021).

5) Maybe the authors can add a bit more details in regard to mitochondrial dysfunction and AD: is the dysfunction also occuring in prodromal AD/ MCI? Is it similar in early or late onset AD? Is mitochondrial failure restricted to neurons or also occurs in glial cells?

These topics have recently been covered in an excellent review by Sousa, Moreira and Cardoso (Biomedicines, 2023). We added a sentence (lines 85-87) and refer to this review for further detail.

6) The authors state that glucose is the main source of energy for the brain. Potentially, they could at least shortly also mention the lactate-shuttle between astrocytes and neurons (e.g., Magistretti and allaman, Nat Rev Neurosci 2018)?

We have added a sentence mentioning the lactate-shuttle between astrocytes and neurons (lines 98-100).

7) I would ask the authors to be a bit more careful about stating that e.g. Ginkgo biloba extract has proven ineffectiv. There are some recent metaanlayses that hint at a positive effect (Xie et al., Cells 2022, Fan et al., JAD 2022). The same might be true for the other mentioned compounds. I would ask to maybe give a more balanced view on this.

We revised the statement on Ginkgo biloba and added the references (lines 139-143).

8) The authors describe very well what happens on the functional level with mitochondria in hibernation. Is there anything known about number of mitochondria or fusion/ fission behavior?

We have added a sentence on fusion and fission as potential regulators of metabolism during hibernation (lines 196-198).

Reviewer 3 Report

Comments and Suggestions for Authors

The authors wrote a novel review comparing the decreased mitochondrial function that occurs in Alzheimer’s disease (AD) neurons to that which occurs during hibernation.  Unlike in AD, the decreased mitochondrial function during hibernation is completely reversible. So, it may be therapeutically useful in AD to mimic the mechanisms of hibernation-induced mitochondrial recovery to attempt to restore mitochondrial function. The review was well-organized, unique, and fills a gap in the literature. The only major omission I found was a lack of description of how the SUL-138 compound that restores mitochondrial electron transport chain function was identified and first tested. The review makes a nice contribution to the literature when new ideas for therapies for AD are desperately needed.

Major comment: Can you further describe in a paragraph the molecular mechanisms through which SUL-138 increases mitochondrial complex I and IV activity and how this compound was discovered? This may allow for other labs to find similar compounds.

Minor comments – wording and grammar

Line 13: cure -> therapy

Line 54: ranging -> ranging from

Line 90: highest -> most

Line 97: phosphorylation complex-I and -III-> phosphorylation, produced mostly be complexes I and III

Line 100: TCA -> TCA cycle

Line 104: cytochrome c -> cytochrome c oxidase (complex IV)

Line 110: complex -> complexes

Line 114: inefficient proton motive force over -> generation from

Line 141: Curcumin -> curcumin

Line 146: Nicotinamide -> precursors of nicotinamide

Line 152: Please also reference mitochondrial complex V (ATP synthase) inhibitor and curcumin derivative J147 showing promise for the treatment of AD [PMID:  31742554 and PMID: 37674139]

Line 176: an TCA -> a TCA

Line 177: TCA -> TCA cycle

Line 180: inhibit of -> inhibit

Line 189: squirrel -> squirrels

Line 209: mitochondrial -> Mitochondrial

Line 222: TCA -> TCA cycle

Line 222: Remove the words “of these enzymes”

Line 239: C57/6N -> C57BL/6N

Line 261: uncoupler -> uncoupling

Line 262: of BAM15 indeed rely -> that BAM15 indeed relies

Line 263: in an -> in a

Line 263: Italicize C. elegans.

Line 265: Remove the words “could also be relevant for AD”

Line 265: it -> BAM15

Line 280 in -> in the

Line 286: mediated -> mediates

Line 294: ATP -> ATP synthesis

Line 303: sel-12 (113). -> sel-12 (113) and in a 3x-Tg mouse model of AD [PMID: 37904949].

Line 320: through cytochrome c reduction -> by binding Fe2+ in heme.

Line 321: is also -> has also been

Line 324: as -> as a

Line 340 and 342: TCA -> TCA cycle

Line 342: the OXPHOS -> OXPHOS

Line 386: passes -> is permeable through

Line 408: synaptoneurosomes -> synaptosomes

Line 419: leans -> relies

Line 424: proposed -> Proposed

Line 434: Remove the word “arousal”

Comments on the Quality of English Language

There are a moderate amount of English grammar and wording changes required as outlined in the critique.

Author Response

Major comment: Can you further describe in a paragraph the molecular mechanisms through which SUL-138 increases mitochondrial complex I and IV activity and how this compound was discovered? This may allow for other labs to find similar compounds.

We have added a paragraph on SUL compound discovery (lines 374-382). The exact nature of the molecular interaction between SUL-138 and specific protein components of the electron transport chain is still subject of investigation.

Minor comments – wording and grammar

We have changed all, as suggested, except for synaptoneurosomes to synaptosomes. We specifically meant synaptoneurosomes which, in contrast to synaptosomes, represent functional presynaptic (synaptosome) and postsynaptic (neurosome) compartments.

Round 2

Reviewer 1 Report

Comments and Suggestions for Authors

The work has introduced a few amendments that do not change y overall opinion. Mitochondrial bioenergetic impairment is an associated pathogenetic contribution to the development of AD, which is essentially caused by misfolding of non-mitochondrial proteins. The contribution of oxidative stress originated from mitochondrial impairment remains an interesting hypothesis but is not supported by robust experimental evidence. The metabolic adaptive changes during torpor in and arousal from hibernation, including stimulation of OXPHOS activities, are interesting clues for general biology but I am difficult to understand the relation with the pathogenesis of AD and the transferability of this information to the understanding of the mechanisms and development of potential therapeutic approaches in AD.